# Ovarian Malignancies Frequency in the Female Population from the Bryansk Region Living in Conditions of Radioactive, Chemical and Combine Contamination (2000–2020)

**DOI:** 10.3390/life11111272

**Published:** 2021-11-21

**Authors:** Anton V. Korsakov, Alexandra A. Golovleva, Vladislav P. Troshin, Dmitry G. Lagerev, Leonid I. Pugach

**Affiliations:** Laboratory “Human Ecology and Data Analysis in the Technosphere”, Bryansk State Technical University, 241035 Bryansk, Russia; aleksgolovleva@yandex.ru (A.A.G.); vptbr32@mail.ru (V.P.T.); LagerevDG@mail.ru (D.G.L.); lip58@mail.ru (L.I.P.)

**Keywords:** environmental pollution, environmental assessment, environmental health, Chernobyl accident, radioactive contamination, chemical pollution, combined contamination, Cesium-137, Strontium-90, average annual effective doses, pollutants, ovarian malignancies, primary morbidity, correlation analysis, regression analysis, Bryansk region

## Abstract

**Background:** Radioactive contamination and chemical pollution of the environment can affect the processes of carcinogenesis, including the formation of malignant neoplasms of the ovaries in women. We used the data of official state statistics for 2000–2020 to test the hypothesis about the effect of radioactive contamination (following the Chernobyl disaster) and chemical pollutants on the incidence of ovarian malignancies in the female population of the Bryansk region. **Methods:** A variety of statistical approaches were used to estimate the incidence of ovarian malignancies, including the Shapiro–Wilk test, Mann–Whitney U test, Spearman’s rank correlation test and linear regression. **Results:** We did not establish statistically significant differences in the frequency of primary morbidity of women with malignant neoplasms of the ovaries, regardless of the environmental conditions of living. Furthermore, no significant correlations were found between the frequency of primary morbidity of ovarian malignancies, both with the level of contamination by Cesium-137 and Strontium-90, and air pollution with volatile organic compounds, carbon monoxide, sulfur dioxide and nitrogen oxides. A statistically significant increase in the long-term trend in the frequency of ovarian malignant neoplasms was revealed in the areas of chemical pollution (*p* = 0.02), however, in other territories, no statistically significant regularities were established. The forecast of the frequency of newly diagnosed malignant neoplasms of the ovaries on average in the Bryansk region shows an increase of 12.4% in 2020 in comparison with the real data for 2020, while the largest increase in predicted values is recorded in the territories of radioactive contamination (by 79.6%), and the least in the combined territories (by 6.9%). **Conclusions:** The results obtained indicate the need for further work to understand the trends in the presence/absence of independent and combined effects of pollutants and the growth of oncogynecological pathology from the perspective of assessing the distant and regional metastasis, histological and immunohistochemical profile of a specific malignant ovarian neoplasm with levels of environmental contamination.

## 1. Introduction

According to the latest estimates of the International Agency for Research on Cancer (IARC) at the World Health Organization (WHO) GLOBOCAN 2020 [1], the incidence of malignant neoplasms in the world has increased to 19.3 million new cases and 10.0 million deaths from them in 2020. Malignant neoplasms of the ovaries occupy the 7th place in terms of prevalence among malignant neoplasms in the world, at the same time as being one of the most fatal oncogynecological pathologies [2]. According to the Russian National Medical Research Center for Oncology, named after N.N. Blokhin, ovarian malignancies in terms of oncological morbidity in Russia is second only to endometrial cancer and cervical cancer in frequency of occurrence [3]. It should be noted that the distribution of patients with ovarian cancer by stages significantly differs from their distribution in cancer of the cervix and uterine body, being characterized by half the proportion of I–II stages among patients with a newly established diagnosis, and a predominance of III–IV stages of the disease. This certainly worsens the prognosis of effective treatment and the quality of life of patients [3].

A number of studies reveal a significant relationship between the risk of malignant neoplasms of the female reproductive system with an increase in the level of technogenic radioactive contamination [4,5,6,7,8,9,10,11,12,13] and chemical pollution [14,15,16,17] of the environment. It is also important that among women living in ecologically unfavorable areas [18], as well as among women who survived the atomic bombing in Hiroshima and Nagasaki [5], the percentage of poorly differentiated ovarian malignancies of a solid structure was higher than that of highly differentiated ones, and the survival rate of women with mucinous cancer was higher than with serous cancer, which indicates the worst prognosis of effective treatment and quality of life in such patients.

As a result of nuclear weapons testing, the disposal of radioactive waste in the seas and major radiation accidents in the second half of the twentieth century (Mayak, 1957; Three-Mile Island, 1979; Chernobyl, 1986), a huge amount of technogenic radionuclides was introduced into the biosphere [19]. At the beginning of the 21st century, this trend continued in connection with a major radiation accident at the Fukushima nuclear power plant on 11 March 2011 (the environment was contaminated with 135 radionuclides, including radioactive daughter products) (IRSN, 2012) [20], the consequences of which will affect for many decades, if not centuries, both the health of the population [21,22] and the environment [23,24]. 

Thirty-five years after the Chernobyl accident, about 5 million people live in the radioactively contaminated territories of Ukraine, Belarus and Russia [12], and the density of radioactive contamination, determined mainly by long-lived ^137^Cs and ^90^Sr, will remain radiologically significant for several decades if not centuries more [12,25,26].

Currently, about 316,000 people in 749 settlements still live in the radioactively contaminated areas of the Bryansk region (Decree of the Government of the Russian Federation of 10.08.2015 No. 1074) [27].

Regular radioecological monitoring in the area of the Bryansk region indicates that the density of soil contamination by ^137^Cs and ^90^Sr in the southwestern territories significantly exceeds the established radiological limits [28], while the accumulated effective radiation doses of the population 35 years after the Chernobyl accident vary by range from units (from 4) to hundreds (up to 299) of mSv [29].

According to official data (On the State and Environmental Protection of the Russian Federation in 2019) [30], in recent years in the Bryansk region there has been an increase in the emission of air pollutants, mostly volatile organic compounds (VOCs) and nitrogen oxides (NO_x_), as well as solid household and industrial waste.

It should be noted that in some areas of the Bryansk region, the population is exposed to the combined influence of radioactive and chemical contamination of the environment [31,32]. Thus, in [33] it was found that the combination of both radioactive and chemical pollutants led to significantly higher frequencies of multiple congenital malformations when compared to regions with only one pollutant (radiation alone: 2.2 times, *p* = 0.034; chemical pollutants alone: 1.9 times, *p* = 0.008). These findings suggest additive and potentially synergistic effects of radioactive and chemical pollutants on the frequencies of multiple congenital malformations.

Accordingly, the increase in the rate of the mutational process, which occurs as a result of environmental pollution and the degradation of the ecological situation, creates a threat to the genetic safety of all living things [34].

In this regard, the study of the health status of the female population living in ecologically unfavorable conditions is highly relevant. Therefore, we carried out a comparative assessment of the frequency of malignant neoplasms of the ovaries in the female population of the Bryansk region living in conditions of radioactive, chemical and combined contamination during 2000–2020.

## 2. Methods

We conducted an ecological and hygienic assessment of the state of the environment and the level of primary morbidity of the female population with malignant neoplasms of the ovaries in the Bryansk region, namely, in 4 cities and 27 districts in terms of radiation (as a result of the Chernobyl accident), chemical (due to atmospheric air pollution) and combined radiation and chemical contamination over a twenty-year period (2000–2019).

The density of radioactive contamination of the territories by ^137^Cs and ^90^Sr due to the Chernobyl accident was estimated according to the data [28], and chemical pollution was acquired from reports on emissions of chemicals into the atmosphere from stationary sources [35]. We identified main gaseous air pollutants: carbon monoxide, sulfur dioxide, nitrogen oxides and VOCs (including benz (a) pyrene, benzene, styrene, pyridine, vinyl chloride, formaldehyde, acrolein and phenol). Recalculation of the amount of gross emissions of chemicals into the atmosphere (tons/year) per city or district area (km^2^) was carried out in (grams/m^2^), according to [35].

According to the information guide [36], we used the average annual effective dose for the population from the Chernobyl component. In addition, the exposure dose rate of gamma radiation (level of natural background radiation) in all uncontaminated areas of the Bryansk region does not exceed 0.20 μSv/h, while in radiation-contaminated areas it often exceeds 0.30 μSv/h and in some settlements the exclusion and resettlement zone values reach 0.8–1.6 μSv/h [36].

The primary morbidity of the female population ovaries malignancies (age 18 and over) in the Bryansk region was analyzed according to the data of the Bryansk regional oncological dispensary [37].

In total, in the Bryansk region during 2000–2020 there were 2647 registered cases of malignant neoplasms of the ovaries in woman, including 439 cases in ecologically safe areas, 1750 cases of chemical pollution and 169 and 289 cases of radioactive and combined contamination, respectively. The recalculation of newly diagnosed malignant neoplasms of the ovaries (absolute values) was carried out per 100,000, taking into account the female population in cities and districts.

Statistical analysis of the data obtained was carried out using the tools of the Stata SE 14.2 package (Stata Corp., College Station, TX, USA). The sample mean (M) and the standard error of the mean (m) were used for estimation of main parameters. The normal distribution of the level of chemical and radioactive contamination was assessed using the Shapiro–Wilk test. We showed that the sample is far from normal distribution both for ^137^Cs and ^90^Sr, and separately for each pollutant and for the sum of pollutants. Therefore, to assess the relationship between the level of chemical and radioactive contamination with the frequency of ovarian malignant neoplasms, we used the Spearman rank correlation test. To test the statistical significance of differences (paired comparisons), we used the Mann–Whitney U test [38].

We calculated the linear regression of the frequency of malignant neoplasms of the ovaries in ecologically different areas in the Bryansk region for 2000–2019. When testing the hypothesis about the relationship between the frequency of malignant neoplasms of the ovaries and the year, the Spearman rank correlation test was used. Calculations of 95% confidence intervals were completed for the angular coefficient *a*, showing the direction of the trend.

Based on the available data, we calculated the prognosis of the frequency of ovarian malignant neoplasms. To do this, we found a linear function *y = ax + b* by the least squares method, which most accurately approximates the available statistical data for each of the indicated categories. We used data for 2000–2019. Using this linear function, we calculated the forecast for 2020 and compared the predicted values with the real ones. The presented forecast will allow us to assess how the real values of the frequency of malignant neoplasms of the ovaries differ from those predicted in the context of the COVID-19 pandemic.

## 3. Results

As a result of the ecological and hygienic analysis of the condition of the environment in cities and districts of the Bryansk region over a twenty-year study period (2000–2019), we ranked the territories (Table 1) depending on the level of chemical pollution of the atmospheric air by the amount of gross emissions of gaseous pollutants (VOCs, SO_2_, CO and NO_x_) on the area of the district, the density of radioactive contamination by ^137^Cs and ^90^Sr due to the Chernobyl accident and the primary morbidity of ovarian malignancies in women. We also presented the average sample sizes for the female population over 18 years old in cities and districts of the Bryansk region (Table 1).

We identified four groups of territories of the Bryansk region according to the degree of unfavorable ecology of the environment (Table 1): (1) ecologically safe areas; (2) areas of chemical pollution; (3) areas of radioactive contamination; (4) areas of combined radiation and chemical contamination.

As Table 1 indicates, the data on the density of radioactive contamination by ^137^Cs and ^90^Sr and the level of chemical pollution by leading gaseous pollutants vary within wide limits. For ^137^Cs—from 4.4 to 460.6 kBq/m^2^, for ^90^Sr—from 0.4 to 16.3 kBq/m^2^. In terms of gross emissions of gaseous pollutants into the air per area (g/m^2^)—from 12 to 32,191, of which: carbon monoxide—from 7 to 5217, nitrogen oxides—from 6 to 10,886, sulfur dioxide—from 0 to 2617 and VOCs—from 0 to 13,470.

Thus, in the group of ecologically safe areas, the density of radioactive contamination is much lower than the established standards for both ^137^Cs (up to 37 kBq/m^2^) and ^90^Sr (up to 5.6 kBq/m^2^). The total level of chemical pollution of atmospheric air by gaseous pollutants is from 12 to 128 g/m^2^, which makes it possible to classify these territories as control (ecologically safe) areas. The frequency of primary morbidity with malignant neoplasms of the ovaries in ecologically safe areas ranges from 15.1 to 26.3; the average over a twenty-year period was 20.6, which is 8.8% less than the all-Russian public values (Table 1).

In the areas of chemical pollution, the gross emissions of gaseous pollutants per area of the district significantly exceed the analogous indicators of ecologically safe territories (sometimes by a factor of thousands), fluctuating within wide limits—from 123 to 32,191 g/m^2^. At the same time, the density of ^137^Cs radioactive contamination varies from 4.4 to 38.4 kBq/m^2^, and ^90^Sr from 0.4 to 5.9 kBq/m^2^. The data obtained allowed us to classify this group of areas as areas of chemical pollution. The incidence rate of malignant neoplasms of the ovaries varies in this group of districts from 12.7 to 29.3, the average value is 22.7, which exceeds the values in ecologically safe regions by 10.2% and practically coincides with the all-Russian values (22.6).

In the group of areas of radioactive contamination, the ^137^Cs contamination density exceeds the established standards by 3.8–12.4 times, and ranges from 139.6 to 460.6 kBq/m^2^. The density of ^90^Sr contamination reaches its maximum values in the Zlynkovsky district (16.3 kBq/m^2^), which exceeds the established standards by 2.9 times, but in two districts (Gordeevsky and Klintsovsky) it does not exceed the established standards, amounting to 5.0 and 4.7 kBq/m^2^. At the same time, the level of atmospheric air pollution by technogenic pollutants is quite low and is comparable to the indicators of ecologically safe areas, ranging from 16.0 to 169 g/m^2^. Such indicators make it possible to classify this group of regions as territories of radioactive contamination. It should be noted that in the areas of radioactive contamination, the incidence of malignant neoplasms of the ovaries ranges from 11.0 to 21.2, the average value is 18.3, which is less than the indicators of the control territories by 11.2% and the all-Russian values by 19.0%.

In the areas of combined contamination, the density of radioactive contamination by ^137^Cs, as well as in the radiation-contaminated territories, exceeds the established standards (by 1.23–12.3 times), amounting to 45.4–456.5 kBq/m^2^. The highest density of ^137^Cs contamination is recorded in the city of Novozybkov (456.5 kBq/m^2^), and the density of ^90^Sr contamination is exceeded only in the city of Novozybkov (9.7 kBq/m^2^). At the same time, in addition to the increased and high level of radioactive contamination, the level of chemical pollution by gaseous pollutants is 2.6–491 times higher than the values of radiation-contaminated areas, amounting to 392–7422 g/m^2^, which allows them to be classified as combined (Table 1). In conditions of combined contamination, the incidence of malignant neoplasms of the ovaries varies from 17.4 to 24.0; the average value was 20.1, which is 2.4% less than the indicators of the control areas and 11.1% less than the all-Russian values.

The results obtained indicate that there are no statistically significant differences in the incidence of malignant neoplasms of the ovaries in cities and districts of the Bryansk region, regardless of the environmental conditions of residence (*p*-values according to the Mann–Whitney U test vary from 0.11 to 0.94), see Table 1.

It should be noted that 35 years after the Chernobyl accident, the average annual effective dose from the Chernobyl component in settlements in the group of ecologically safe areas and areas of chemical pollution does not exceed 0.3 mSv per year, while in the group of radioactive and combined contamination the maximum values reach 5.5 mSv per year [35].

The dynamics of newly diagnosed ovarian malignancies (absolute values) in ecologically different territories of the Bryansk region in 2000–2020 are presented in Table 2. The data in Table 2 indicate that the number of malignant neoplasms of the ovaries in areas of chemical pollution ranges from 34 to 111 cases per year, in areas of radioactive contamination from 2 to 17, in areas of combined contamination from 8 to 20 and in ecologically safe areas from 12 to 32.

The dynamics of the frequency of primary morbidity with ovarian malignancies in ecologically different territories of the Bryansk region in 2000–2020 are presented in Table 3. The data in Table 3 indicate that the frequency of malignant neoplasms of the ovaries differs from the absolute values. So, in areas of chemical pollution it ranges from 9.1 to 31.0 per year, in areas of radioactive contamination from 4.1 to 43.5, in areas of combined contamination from 11.3 to 28.4 and in ecologically safe areas from 11.2 to 34.4 per 100,000.

Since the dynamics of the frequency of malignant neoplasms of the ovaries in 2000–2019 in ecologically different territories of the Bryansk region often have a spasmodic character over the years, we performed a linear calculation for three-year periods (2000–2002, 2003–2005, 2006–2008, 2009–2011, 2012–2014, 2015–2017) and over the last 2 years (2018–2019)—Figure 1. We revealed an increase in the long-term trend of the frequency of malignant neoplasms of the ovaries in ecologically safe areas and in areas of chemical pollution and radioactive contamination, and a slight decrease in areas of combined contamination. However, a statistically significant increase was found only in the group of territories of chemical pollution (*p* = 0.02)—Figure 1.

Our forecast of the frequency (based on 2000–2019 data) of newly diagnosed malignant ovarian neoplasms on average in all cities and districts of the Bryansk region shows an increase of 12.4% in 2020 in comparison with real data for 2020. The reason for this, in all probability, is a reorientation of the healthcare system in connection with the COVID-19 pandemic—Figure 1.

Furthermore, the increase in predicted values in comparison with real data is uneven. Thus, the greatest increase in newly diagnosed malignant neoplasms of the ovaries was recorded in areas of radioactive contamination by 79.6% (28.2 forecast for 2020, 15.7 real values for 2020), and a less pronounced increase was found in ecologically safe areas by 18.8% (forecast 25.3, real result 21.3), in areas of chemical pollution by 11.9% (30.2 versus 27.0) and combined contamination by 6.9% (19.9 versus 18, 6)—Figure 1.

Correlation analysis of the primary incidence of ovarian malignant neoplasms in cities and districts of the Bryansk region with the level of radiation contamination and chemical pollution of the environment (Table 4) did not reveal significant links between the incidence of primary ovarian malignant neoplasms as with the level of ^137^Cs contamination (ρ = −0.19, *p* = 0.31) and ^90^Sr (ρ = 0.02, *p* = 0.92), as well as air pollution by VOCs (ρ = 0.19, *p* = 0.32), CO (ρ = 0.09, *p* = 0.61), NO_x_ (ρ = 0.22, *p* = 0.23) and SO_2_ (ρ = 0.27, *p* = 0.14).

## 4. Discussion

As a result of our study, we did not reveal an increased frequency of ovarian malignancies in ecologically unfavorable areas in comparison with ecologically safe ones, as well as a relationship between the level of chemical and radioactive contamination with the primary incidence of ovarian malignant neoplasms. This indicates that the effect of endogenous factors on female reproductive processes is of a greater extent than exogenous.

It should be noted that among the circumstances of the risk of malignant neoplasms, there are many exogenous and endogenous factors, which are practically impossible to take into account. According to the literature [39,40,41,42,43,44], IARC and WHO [45,46], among the main risk factors for malignant neoplasms (including malignant neoplasms of the female reproductive system) are the use of tobacco, alcohol, unhealthy diet, physical inactivity, overweight, drug therapy for infertility, hereditary predisposition, chemical (polycyclic aromatic hydrocarbons, dioxins, pesticides, aflatoxins, arsenic, formaldehyde, nickel, asbestos, cadmium and many others), physical (ionizing and ultraviolet radiation) and biological (infections caused by viruses, bacteria or parasites) environmental carcinogens. Some have suggested that the upward trend in the incidence of malignant neoplasms in the world may reflect some general trends in the increase in the genetic load in human populations, due to the growth of chemical pollution and radiation contamination of the biosphere by “eternal” (half-lives of which are more than a hundred years) and “global” (rapidly spreading from the place of pollution throughout the biosphere) pollutants [34].

When conducting further research, it is necessary to utilize:More exact measurements of radioactive contamination and chemical pollution of the environment;More exact estimates of accumulated radiation doses in the population (primarily ^137^Cs and ^90^Sr);More exact analysis of the distribution of sources of air pollution and the deposition of emissions from chemicals, taking into account meteorological factors;A more complete analysis of the dynamics of malignant neoplasms of the ovaries in women of different age groups;An assessment of the economic and social situation in the cities and districts of the Bryansk region (average wages, retail trade turnover, consumer price index, production index, mortality and natural increase, birthrate).

## 5. Conclusions

We did not find statistically significant differences in the frequency of primary morbidity with malignant neoplasms of the ovaries in women, regardless of the environmental conditions of residence.We did not find significant correlations between the frequency of primary morbidity of malignant neoplasms of the ovaries, both with the level of contamination by ^137^Cs and ^90^Sr, and air pollution with volatile organic compounds, carbon monoxide, sulfur dioxide and nitrogen oxides.We found a significant increase in the long-term trend in the frequency of malignant neoplasms of the ovaries in areas of chemical pollution (*p* = 0.02), however, in other areas, no statistically significant regularities were established.Our forecast for the frequency of newly diagnosed malignant neoplasms of the ovaries on average in the Bryansk region showed an increase of 12.4% in 2020 in comparison with real data for 2020, while the largest increase in predicted values was recorded in the territories of radioactive contamination (by 79.6%), and the least in the combined territories (by 6.9%).The results obtained indicate the need for further work to understand the trends in the presence/absence of independent and combined effects of pollutants on the growth of oncogynecological pathology from the perspective of assessing distant and regional metastasis, histological and immunohistochemical profile of a specific malignant ovarian neoplasm with levels of environmental contamination.

## Figures and Tables

**Figure 1 life-11-01272-f001:**
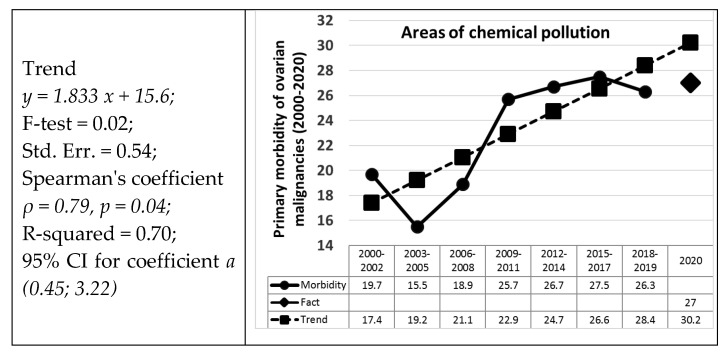
Dynamics of the frequency of primary morbidity ovarian malignancies in ecologically different territories of the Bryansk region with long-term trend lines for three years in the period 2000–2019 and forecast for 2020 (per 100,000).

**Table 1 life-11-01272-t001:** Ranking of areas within the Bryansk region by the level of radiation, chemical and combined environmental contamination and the frequency of primary morbidity of ovarian malignancies in the female population (2000–2019).

№	Cities and Districts of the Bryansk Region	Main Gaseous Air Pollutants					Contamination Density, kBq/m^2^		Primary Morbidity of Ovarian Malignancies (per 100,000), M ± m
		Total	Of Them:						
			VOCs	NO_x_	SO_2_	CO	^137^Cs	^90^Sr	
		Gross Emissions of Gaseous Pollutants Per Area, g/m^2^							
**Ecologically safe areas (control)**									
1	Rognedinsky *(n = 3416)*	12	0	6	0	7	21.7	0.8	25.8 ± 6.5
	Suzemsky *(n = 7845)*	27	5	9	1	13	18.6	2.5	26.3 ± 4.5
	Mglinsky *(n = 8698)*	31	6	6	2	17	6.6	0.6	17.5 ± 3.1
	Kletnyansky *(n = 8990)*	47	27	5	5	10	5.4	0.5	21.6 ± 3.5
	Navlinsky *(n = 12,468)*	53	12	13	4	25	18.9	0.8	18.7 ± 3.6
	Dubrovsky *(n = 9087)*	56	13	17	0.4	26	7.2	0.4	19.4 ± 5.0
	Brasovsky *(n = 9423)*	64	10	19	6	29	25.2	0.4	20.0 ± 3.2
	Sevsky *(n = 7581)*	68	20	10	24	14	18.9	1.4	19.4 ± 4.5
	Komarichsky *(n = 8086)*	99	25	19	9	46	27.1	1.0	15.1 ± 2.9
	Karachevsky *(n = 16,442)*	115	29	34	1	51	13.9	0.8	25.6 ± 2.7
	Surazhsky *(n = 10,894)*	128	35	35	6	52	8.2	0.4	17.9 ± 3.2
	**Average value**	**63.2**	**16.1**	**15.7**	**5.3**	**26.4**	**15.6**	**0.9**	**20.6 ± 1.3** **(−2.0 *)**
**Areas of chemical pollution**									
2	Pogarsky *(n= 13,398)*	123	65	22	4	32	29.9	1.1	29.3 ± 3.9
	Zhiryatinsky *(n= 3274)*	155	104	16	1	35	5.4	0.8	18.1 ± 4.5
	Zhukovsky *(n= 16,428)*	196	22	53	40	80	6.6	0.8	19.9 ± 1.7
	Trubchevsky *(n = 16,659)*	275	88	27	2	158	23.6	0.8	16.0 ± 2.5
	Pochepsky *(n = 18,827)*	364	223	33	3	106	5.4	0.5	19.5 ± 3.2
	Unechsky *(n= 18,519)*	559	292	58	32	177	7.2	0.8	24.1 ± 3.1
	Vygonichsky *(n = 9155)*	857	749	37	2	70	9.5	0.4	12.7 ± 3.9
	Bryansky *(n = 24,737)*	959	813	47	13	86	5.7	0.4	23.8 ± 1.9
	Town Seltso *(n= 8140)*	5208	773	2405	97	1934	4.4	0.8	23.4 ± 2.7
	Dyatkovsky *(n = 33,907)*	8045	339	3760	1139	2807	38.4	1.1	22.0 ± 1.8
	City Bryansk *(n = 202,954)*	32,191	5217	10,886	2617	13,470	8.8	5.9	23.7 ± 1.5
	**Average value**	**4450.9**	**792.1**	**1576.7**	**359.1**	**1723.2**	**13.2**	**1.2**	**22.7 ± 1.2** **(+0.1 *)**
**Areas of radioactive contamination**									
3	Krasnogorsky *(n = 6273)*	16	1	5	0	9	303.4	9.3	18.8 ± 4.2
	Gordeevsky *(n = 5197)*	29	2	11	0.2	15	328.6	5.0	11.0 ± 4.2
	Zlynkovsky *(n= 5654)*	37	5	11	4	18	412.4	16.3	18.3 ± 3.7
	Novozybkovsky *(n = 5558)*	51	10	0	0	41	460.6	8.4	14.6 ± 5.7
	Klimovsky *(n= 13,731)*	72	16	8	15	33	139.6	6.4	21.2 ± 3.6
	Klintsovsky *(n = 8920)*	169	17	70	2	80	194.4	4.7	20.0 ± 3.1
	**Average value**	**62.3**	**8.5**	**17.5**	**3.5**	**32.7**	**306.5**	**8.4**	**18.3 ± 2.0** **(−4.3 *)**
**Areas of combined radiation-chemical contamination**									
4	Starodubsky *(n = 18,247)*	392	316	24	9	43	45.4	1.4	20.9 ± 2.4
	City Klintsy *(n = 32,128)*	7264	2059	2616	139	2450	195.6	3.0	17.4 ± 1.1
	Sity Novozybkov *(n = 18,294)*	7422	1778	2159	406	3079	456.5	9.7	24.0 ± 2.1
	**Average value**	**5026**	**1384.3**	**1599.7**	**184.7**	**1857.3**	**232.5**	**4.7**	**20.1 ± 0.9** **(−2.5 *)**

*Note *.* Difference from the all-Russian primary morbidity of ovarian malignancies (2000–2019). Significance level while checking the hypothesis about differences in the frequency of primary morbidity of ovarian malignancies according to the Mann–Whitney U test on ecologically safe areas and areas of the chemical (*p* = 0.67), radioactive (*p* = 0.21) and combined (*p* = 0.94) contamination; chemical and radioactive (*p* = 0.11), chemical and combined (*p* = 0.94), radioactive and combined (*p* = 0.30) contamination. *n* = average sample size by female population over 18 years old. Bold: the average sample sizes for the female population over 18 years old in cities and districts of the Bryansk region.

**Table 2 life-11-01272-t002:** Dynamics of newly identified ovarian malignancies (absolute values) in ecologically different territories of the Bryansk region in 2000–2020.

Years	Territories *			
	*CP*	*RC*	CC	*ES*
2000	72	8	12	24
2001	74	11	20	19
2002	75	12	15	30
2003	82	5	17	25
2004	58	4	8	18
2005	34	2	16	13
2006	76	7	11	12
2007	56	5	13	15
2008	79	8	13	16
2009	104	12	16	20
2010	87	8	14	17
2011	94	10	14	21
2012	80	7	17	26
2013	111	6	9	23
2014	100	3	11	23
2015	111	11	17	22
2016	85	11	13	24
2017	96	10	14	32
2018	84	17	13	20
2019	99	6	13	20
2020	93	6	13	19

*** Territories: CP—chemical pollution; RC—radioactive contamination; CC—combined contamination; ES—ecologically safe.

**Table 3 life-11-01272-t003:** Dynamics of the frequency of primary morbidity ovarian malignancies in ecologically different territories of the Bryansk region in 2000–2020 (per 100,000).

Years	Territories *			
	*CP*	*RC*	CC	*ES*
2000	19.3	15.0	16.9	21.2
2001	19.8	21.0	28.4	17.0
2002	20.0	23.5	21.3	27.1
2003	21.9	10.0	24.0	22.7
2004	15.5	8.2	11.3	16.5
2005	9.1	4.1	22.6	12.0
2006	20.4	14.7	15.7	11.2
2007	15.1	10.7	18.6	14.1
2008	21.3	17.3	18.6	15.1
2009	28.0	26.3	23.0	19.0
2010	23.5	17.7	20.3	16.2
2011	25.5	22.5	20.6	20.5
2012	21.8	16.1	25.3	25.7
2013	30.5	14.2	13.4	23.0
2014	27.7	7.1	16.4	23.5
2015	31.0	26.9	25.4	22.9
2016	24.2	27.3	19.5	25.4
2017	27.2	25.2	21.1	34.4
2018	24.0	43.5	19.7	21.9
2019	28.5	15.6	19.9	22.3
2020	27.0	15.7	18.6	21.3

*** Territories: CP—chemical pollution; RC—radioactive contamination; CC—combined contamination; ES—ecologically safe.

**Table 4 life-11-01272-t004:** Correlation analysis of the primary morbidity of ovarian malignancies of the female population in cities and districts of the Bryansk region with the level of radiation and chemical contamination of the environment (2000–2019).

Cities and Districts of the Bryansk Region	Main Gaseous Air Pollutants					Contamination Density kBq/m^2^		Primary Morbidity of Ovarian Malignancies (per 100,000)
	Total	Of Them:						
		VOCs	NO_x_	SO_2_	CO	^137^Cs	^90^Sr	
	Gross Emissions of Gaseous Pollutants Per Area, g/m^2^							
Rognedinsky	12	0	6	0	7	21.7	0.8	25.8
Suzemsky	27	5	9	1	13	18.6	2.5	26.3
Mglinsky	31	6	6	2	17	6.6	0.6	17.5
Kletnyansky	47	27	5	5	10	5.4	0.5	21.6
Navlinsky	53	12	13	4	25	18.9	0.8	18.7
Dubrovsky	56	13	17	0.4	26	7.2	0.4	19.4
Brasovsky	64	10	19	6	29	25.2	0.4	20.0
Sevsky	68	20	10	24	14	18.9	1.4	19.4
Komarichsky	99	25	19	9	46	27.1	1.0	15.1
Karachevsky	115	29	34	1	51	13.9	0.8	25.6
Surazhsky	128	35	35	6	52	8.2	0.4	17.9
Pogarsky	123	65	22	4	32	29.9	1.1	29.3
Zhiryatinsky	155	104	16	1	35	5.4	0.8	18.1
Zhukovsky	196	22	53	40	80	6.6	0.8	19.9
Trubchevsky	275	88	27	2	158	23.6	0.8	16.0
Pochepsky	364	223	33	3	106	5.4	0.5	19.5
Unechsky	559	292	58	32	177	7.2	0.8	24.1
Vygonichsky	857	749	37	2	70	9.5	0.4	12.7
Bryansky	959	813	47	13	86	5.7	0.4	23.8
Town Seltso	5208	773	2405	97	1934	4.4	0.8	23.4
Dyatkovsky	8045	339	3760	1139	2807	38.4	1.1	22.0
City Bryansk	32,191	5217	10,886	2617	13,470	8.8	5.9	23.7
Krasnogorsky	16	1	5	0	9	303.4	9.3	18.8
Gordeevsky	29	2	11	0.2	15	328.6	5.0	11.0
Zlynkovsky	37	5	11	4	18	412.4	16.3	18.3
Novozybkovsky	51	10	0	0	41	460.6	8.4	14.6
Klimovsky	72	16	8	15	33	139.6	6.4	21.2
Klintsovsky	169	17	70	2	80	194.4	4.7	20.0
Starodubsky	392	316	24	9	43	45.4	1.4	20.9
City Klintsy	7264	2059	2616	139	2450	195.6	3.0	17.4
Sity Novozybkov	7422	1778	2159	406	3079	456.5	9.7	24.0
Correlation coefficients *(ρ)* and levels of their statistical significance *(p)*								
-	**ρ = 0.17** ***p* = 0.37**	**ρ = 0.19** ***p* = 0.32**	**ρ = 0.22** ***p* = 0.23**	**ρ = 0.27** ***p* = 0.14**	**ρ = 0.09** ***p* = 0.61**	**ρ = −0.19** ***p* = 0.31**	**ρ = 0.02** ***p* = 0.92**	**-**

## Data Availability

We used the data of official state statistics by the incidence of ovaries malignancies in woman for 2000–2020 according to the data of the Bryansk regional oncological dispensary.

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
