# Peer review of "Ovarian Malignancies Frequency in the Female Population from the Bryansk Region Living in Conditions of Radioactive, Chemical and Combine Contamination (2000–2020)"

_life, 2021, doi:10.3390/life11111272_

Round 1

Reviewer 1 Report

General comment

This is an ambitious study that attempted to examine the risks of mixed exposure to chemicals and radiation using an ecological study and authors claim that further verification is required.

However, it appears that comparisons between regions are being made using crude indicators. If this is the case, and if the age distribution of the population differs between regions, it is strongly recommended to improve the quality of the indicators evaluated considering the age distributions.

Specific comments

Abstract

Authors stated the hypothesis and described the data used it the background. In a structured abstract, the hypothesis should be presented in the objective, and the research design and data used should be clearly stated in the methods. Statistical test methods may be abbreviated. The first word of the first sentence of the results should be capitalized (same for the conclusion). The conclusion should present the answer for the hypothesis presented in the objective.

Ethical consideration

Although open data was used in this study, it is necessary to mention research ethics.

Line 56

“as well as among women why survived the atomic bombing “

Make sure the "why" is correct.

Line 68 and 73

“if not centuries”

Do you mean, not hundreds of years?

Line 72

“Cs-137 and Sr-90 ”

The expression of radioisotopes is not consistent throughout this paper comparing such as line 77.

Line 81

“from units to hundreds of mSv”

The lower limit of the range must be clearly indicated.

Line 113

“amount of gross emissions of chemicals into the atmosphere (tons/year per km2) was converted to g/m2 according [9].“

A reference is indicated, but the method is not understandable since it is written in Russian, so it is necessary to specify what concept was used to convert “the mass per unit area per year” to “the mass per unit area”.

Line 122

“The primary morbidity of the female population with malignant neoplasms of the ovaries (age from 18 years) in the Bryansk region was analyzed according to the data of the Bryansk regional oncological dispensary [26] ”

The meaning of "primary" in this context needs to be clarified. A reference is indicated, but the method is not understandable since it is written in Russian, so the method of age adjustment also needs to be clearly stated.

Line 125

Several results are described in Methods.

Line 126

“ecologically safe areas”

What criteria were used to select the "ecologically safe areas" shown in Table 1?

Line 131

“The sample mean (M) and the standard error of the mean (m) were used as the mean.“

The exact same sentence is duplicated.

The reviewer thinks “standard error of the mean (m)” shows the standard deviation of the estimated mean not “as the mean”.

Line135

“He showed that the sample is far from normal distribution”

It may be useful to show the distribution in a histogram.

Line 139

“paired comparisons”

The reviewer believes that the Mann-Whitney U-test may have been used because normality cannot be assumed for two unrelated groups.

Line 145

“the coefficient A ”

Show us what "A" stands for.

“To do this, we found a linear function y=ax+b by the least squares method,”

State the validity of applying linear regression in the discussion since a R-squared was 0.002.

Line 242

“presented in Table. 3. ”

There is an extra period after “Table 3”.

Line 249

“(per 100,000). ”

Morbidity is in person-years.

Figure 1

The units of the vertical axis need to be indicated.

Author Statement

In this study, it seems that no experiments were conducted.

Refences

The format needs to be aligned.

Underlines are not necessary for references such as 17,23,28,29,40,42.

Author Response

We are sending a revised article in which the authors tried to take into account all comments without exception. We express our gratitude to you for a deep constructive analysis, which allowed us to significantly strengthen and improve the text of the manuscript.

You can see in the attached file our answers (marked in red) to your comments and recommendations. The text marked in blue in the manuscript has been changed based on the requirements of the editor after checking for plagiarism.

Reviewer 2 Report

Interesting work using unique data. It would be interesting to do the same analysis using Poisson regression https://en.wikipedia.org/wiki/Poisson_regression

Unlike many research papers and articles on the subject the work is utterly realistic, it neither blows out of proportion, nor distort the collected the effect of radiation on women’s health while honesty stating the facts. Also, I find that the collected data on women's cancer was given a thorough thought and subverted to honest and unbiased analysis.

p. 8, rows 131 – 133: the phrase is duplicated.

Author Response

We are sending a revised article in which the authors tried to take into account all comments without exception. We express our gratitude to you for a deep constructive analysis, which allowed us to significantly strengthen and improve the text of the manuscript.

You can see our answers in the attached file (marked in red) to your comments and recommendations. The text marked in blue in the manuscript has been changed based on the requirements of the editor after checking for plagiarism.
